

# A Comparison of the Discrete Cosine and Wavelet Transforms for Hydrologic Model Input Data Reduction

Wright Ashley[1], Walker Jeffrey P.[1], Robertson David E.[2], and Pauwels Valentijn R.N.[1]

[1]Department of Civil Engineering, Monash University, Clayton, Victoria, Australia.
[2]CSIRO, Land and Water, Clayton, Victoria, Australia

*Correspondence to:* Ashley Wright (ashley.wright@monash.edu)

**Abstract.** The treatment of input data uncertainty in hydrologic models is of crucial importance in the analysis, diagnosis and detection of model structural errors. Model input data reduction techniques decrease the dimensionality of input data, thus allowing modern parameter estimation algorithms to more efficiently estimate errors associated with input uncertainty and model structure. The Discrete Cosine Transform (DCT) and Discrete Wavelet Transform (DWT) are used to reduce the dimensionality of rainfall time series observations from the 438 catchments in the MOdel Parameter Estimation eXperiment (MOPEX) data set. The rainfall time signals are then reconstructed and compared to the measured hyetographs using standard simulation performance summary metrics and descriptive statistics as well as peak discharge errors. The results convincingly demonstrate that the DWT is superior to the DCT and best preserves and characterizes the observed rainfall data records. It is recommended that the DWT be used for model input data reduction in hydrology in preference over the DCT.

## 1 Introduction

Rainfall uncertainty is the biggest obstacle hydrologists face in their pursuit of accurate, precise and timely streamflow forecasts (McMillan et al., 2011). Unfortunately, errors in rainfall time series data may lead to hydrological model parameter estimates that produce adequate streamflow simulations during calibration (Beven, 2006). This can lead to poor quality streamflow predictions for independent periods, and low confidence in the ability of streamflow forecasts for short and long term forecasts. Consequently, a precise and accurate representation of rainfall uncertainty is paramount for robust parameter estimation streamflow forecasting and Quantitative Precipitation Forecasts (QPFs). Furthermore, Robertson et al. (2013) and Shrestha et al. (2015) have demonstrated that skill can be added to QPFs by postprocessing with past observations. As such, skill can be added to QPFs and consequently flood forecasts, through developing a greater understanding of rainfall uncertainty.

The propagation of input errors in rainfall runoff modelling impedes the hydrologic community's ability to validate model structural error. Despite the vast amount of literature on rainfall measurement, estimation, statistical analysis (Testik and Gebremichael, 2010) and quality control procedures (World Meteorological Organization, 2008), a shroud of uncertainty still surrounds how rainfall and its associated uncertainty should be addressed in rainfall runoff modeling. The implementation of uncertainty analysis in many hydrological applications is also often limited by computational power.



Recent advancements in computational power as well as remote sensing have led to considerable improvements in data availability and quality (Cloke and Pappenberger, 2009). These improvements can be leveraged to increase the hydrological and flood forecasting knowledge base and consequently provide water policy decision makers and emergency management services with higher quality information. The advancement of computational power has aided the search for optimal model

parameters. These approaches intially focussed on finding only the global optimum values of the parameters for a given objective function (Duan et al., 1994; Gan and Biftu, 1996; Thyer et al., 1999). In the past two decades the interest has switched to the assessment of parameter and prediction uncertainty. Examples of such methods include Bayesian recursive parameter estimation (Thiemann et al., 2001), the limits of acceptability approach (Beven, 2006; Blazkova and Beven, 2009), the BAyesian Total Error Analyis (BATEA) framework (Kavetski et al., 2006a, b; Kuczera et al., 2006; Thyer et al., 2009;

Renard et al., 2011), the Simultaneous optimization and data assimilation (SODA) (Vrugt et al., 2005), the DREAM algorithm and its variations (Vrugt et al., 2005, 2008, 2009a, b; Vrugt and Ter Braak, 2011; Laloy and Vrugt, 2012; Sadegh and Vrugt, 2014), Bayesian model averaging (Butts et al., 2004; Ajami et al., 2007; Vrugt and Robinson, 2007), the hypothetico-inductive data based mechanistic modeling framework of Young (2013) and Bayesian data assimilation (Bulygina and Gupta, 2011). It is through the development of these parameter estimation algorithms that hydrologists are able to explore input uncertainty.

Kavetski et al. (2006b) and Vrugt et al. (2008) identified the need represent true catchment rainfall and its associated uncertainty using parameters, both applied a rainfall multiplier to storm events. The use of a parametric representation of rainfall with an effective sampler has the ability to jointly estimate hydrologic model parameter distributions as well as input uncertainty. As in most hydrological problems there is a lack of sufficient data to obtain a unique solution. Thus, estimating a rainfall multiplier for each time step further complicates the task at hand by making the problem more underdetermined. Data reduction

transformations offer the potential to reduce the dimensionality of the parameter estimation problem to enable a more robust inference. The combination of the use of model input data reduction techniques with parameter estimation algorithms allows links to be explored between rainfall input error, QPF postprocessing algorithms and errors associated with model structure, parameter estimation, and systematic and random errors associated with observations. Signal transforms such as the Discrete Cosine Transform (DCT) and the Discrete Wavelet Transform (DWT) can be used as tools for model input data reduction.

Kumar and Foufoula-Georgiou (1997) introduced wavelet analysis to the geophysical sciences. Schaefli and Zehe (2009) analyzed hydrological model performance and conducted parameter estimation in the wavelet domain and Nalley et al. (2012) used DWTs to analyze rapidly and slowly changing events in both streamflow and precipitation time series. Montanari and Toth (2007) discussed the opportunities calibration of hydrologic models in the spectral domain offers to ungauged basins and Pauwels and De Lannoy (2011) used a discrete Fourier transform to calibrate water and energy balance models in the spectral

domain, whilst De Vleeschouwer and Pauwels (2013) calibrated the rainfall runoff model known as the Probability Distributed Model (PDM) in the spectral domain. To date there have been no instances in which the suitability of different transforms has been assessed for their use as a tool in hydrologic model input data reduction. Further, model input data reduction has not been used to infer hydrologic input uncertainty. Consequently, other fields need to be looked at for examples of model input data reduction. Bruce et al. (2002) have highlighted the efficiency of the DWT to reduce hyperspectral data and automatically

classify ground vegetation from hyperspectral data. Studies in geophysics and hydrogeology have used the DCT (Jafarpour



et al., 2009, 2010; Linde and Vrugt, 2013; Lochbühler et al., 2014, 2015), Discrete Wavelet Transform (Davis and Li, 2011; Jafarpour, 2011) and Karhunen-Loève expansion (Dostert et al., 2006, 2009; Marzouk and Najm, 2009; Laloy et al., 2013) to reduce the 2-D and 3-D parameterization of subsurface structures before model inversion takes places. Amongst many other applications, DCTs and DWTs have been used extensively in image and signal processing.

The motivation for the development of wavelets grew out of limitations of the Fourier transform and it's respective derivations. Fourier transforms use sinusoidal functions to represent the spectral component of an input signal, thus a periodic signal could be represented using a smaller number of Fourier coefficients than the number of input data points. A pitfall of the Fourier transform is that it represents the spectral components of a signal, without any indication of the time localization of those specific spectral components. In order to account for this, the Windowed Fourier Transform (WFT), sometimes referred

to as the short time Fourier transform, segments the signal into discrete time windows before performing the Fourier analysis. A major drawback to this approach is that the uncertainty principle of signal processing imposes a limitation on the time and frequency resolutions that can be obtained for a given signal. As a response to this Daubechies (1990) produced discrete basis functions with good time and frequency localization. In conjunction with the pyramid algorithm, as described by Mallat (1989), This work formed the basis for multi-resolution analysis with the DWT (Polikar, 1999). The pyramid algorithm decomposes

an input signal into high and low frequency components. Labat (2005) has pointed out that Fourier transforms and their derivatives are not well suited to reconstruct hydrologic data, which are generated by transient mechanisms. This is due to the Fourier transforms poor capability to represent sporadic high frequency events when dimensionally reduced. If model input data are to be accepted by the hydrologic community it is of critical important that the transform used is able to reconstruct transient events. Through a comparative study it will be shown that DWTs are a good multi-resolution alternative to the DCT.

Traditionally, transform coefficients are the result of a convolution operation on an input signal. However, the aim of model input data reduction is to estimate these transform coefficients. Hence, they shall be referred to as transform parameters herein. This paper provides a comparison of the DCT and DWT and their ability to be used as a tool for model input data reduction. To address the requirements for hydrologic model input data reduction, this paper addresses (i) theoretical differences between the DCT and DWT, (ii) methodologies to reduce input rainfall to parameters, and (iii) validation of the proposed methodologies

using several simulation performance summary metrics.

## 2   Model Input Data Reduction Theory

For this study, model input data reduction theory is introduced using a lumped conceptual watershed model. Consider a non-linear model $\mathcal{F}(\cdot)$, which simulates $n$ discharge values, $\widehat{\boldsymbol{Y}} = \{\widehat{y}_1, \ldots, \widehat{y}_n\}$ in mm/day according to

$$\widehat{\mathbf{Y}} = \mathcal{F}(\boldsymbol{\theta}, \widetilde{\mathbf{x}_0}, \widehat{\mathbf{E}}, \widehat{\mathbf{R}}), \tag{1}$$

where the model input arguments are the $1 \times d$ vector $\boldsymbol{\theta}$, with arbitrary model parameter values, the $1 \times m$ vector $\widetilde{\boldsymbol{x}_0}$ with

values of the initial states in mm, and the $1 \times n$ vectors $\widehat{\mathbf{E}} = \{\widehat{e}_1, \ldots, \widehat{e}_n\}$ and $\widehat{\mathbf{R}} = \{\widehat{r}_1, \ldots, \widehat{r}_n\}$ store the observed values of the potential evapotranspiration (PET) and rainfall in mm/day, respectively. Note that $\widehat{\mathbf{R}}$ is used to represent rainfall and not





precipitation, as snow, hail and other forms of precipitation are not considered. The ^(hat) symbol is used to denote measured quantities and the ~(tilde) symbol reflects variables that are either reconstructed or could, in theory be observed in the field but due to their conceptual nature are difficult to determine accurately. If the traditional hydrological perspective in which the inputs $\boldsymbol{E}$ and $\boldsymbol{R}$ are considered to be fixed and known quantities is relaxed, and rainfall is now considered unknown, then a new

inference problem arises in which the input rainfall is estimated via treating the input rainfall as a series of parameters. Inference problems in which the input is considered unknown can be dealt with using a Bayesian framework. Such inference problems have been considered by Kavetski et al. (2006a) and Vrugt et al. (2008) but are outside the scope of this paper. Consequently a suitable parametric representation of rainfall must be determined. Given a daily precipitation data record with $n$ observations in mm, n rainfall parameters could be used to represent the input hyetograph. This approach would be particularly elegant and

parsimonious, yet for a 10 year record of daily discharge data, the inference problem would grow from $d$ model parameters to roughly $10 \times 365 + d = 3650 + d$ parameters. These values would need to be estimated from the observed rainfall and discharge data record, respectively. As many hydrological models are already underdetermined the introduction of additional parameters would make the model even less determinable. Additionally an excessive amount of CPU-time is required to solve for a 3600+ dimensional posterior parameter distribution. An alternative approach is therefore necessary. The reduction of model input

data is investigated by representing rainfall data records of the MOdel Parameter Estimation eXperiment (MOPEX) data with a much lower dimensional sparse transform.

Sparse transforms convey large amounts of data using fewer parameters than data points in the initial signal. An input rainfall signal can be reduced to sparse transform parameters. The modification of even one parameter will allow multiple rainfall observations to be altered. Some or all of these transform parameters can be altered before the transform is inverted to produce

a new input signal for streamflow simulation and posterior analysis. The use of sparse transforms to represent input time series enables input uncertainty to be explored. To adequately compare the ability of discrete wavelet and Fourier based transforms, to reduce hydrologic input data to parameters for uncertainty estimation, both theoretical and analytical comparisons are made.

## 2.1 Overview of the DCT and DWT

Wavelet and Fourier transforms are invertible transforms in which a forward convolution operation can be used to decompose

a signal into various components. Similarly, a backwards deconvolution operation can be applied to retrieve the original signal. Fourier based transforms decompose signals into frequency components and are best used for regular time-invariant signals that do not exhibit time specific information. Alternatively, wavelet based transforms decompose signals into frequency and time components. The advantage of using wavelet functions to transform data is that time specific information about when higher frequency components occur can be preserved. To obtain time specific information, Fourier based transforms can be

applied over pre-specified temporal windows. Yet, this approach is limited by the uncertainty principle of signal processing. The uncertainty principle of signal processing imposes a lower limit on obtainable resolutions in the time-frequency domain such that

$$\sigma_t \sigma_\omega \geq \frac{1}{2}, \tag{2}$$





where $\sigma_{\mathrm{t}}$ [s] and $\sigma_{\omega}$ [s$^{-1}$] are the respective temporal and frequency widths used in the sparse transform.

Applying the uncertainty principle of signal processing (Equation 2) it is clear that any attempt to narrow the temporal period analyzed to gain increased resolution in the time domain would be met by a widening of the frequency spectrum, and consequently a loss of resolution in the frequency domain.

5 Considering, there is no time-frequency window that is able to obtain limitless resolution in both the time and frequency domains, it is clear that an alternative solution must be found. Wavelet transforms can be used to decompose a signal into different levels that consist of different time and frequency resolution windows. Thus the wavelet transform is able to be configured to simultaneously obtain high levels of resolution in both the time and frequency domains. For a more detailed discussion on wavelets and sparse transforms the reader is referred to Mallat (2009).

## 2.2 Discrete Cosine Transform

The DCT (Ahmed et al., 1974) is a version of the WFT that has advantageous properties for the field of data compression. Due to the boundary conditions of the cosine function, the DCT is well suited to represent an observed input signal with a minimal number of parameters, in this case rainfall $\widehat{\mathbf{R}}(t)$. The DCT parameters $\mathbf{p}(i)$ are calculated as

$$\mathbf{p}(i) = \mathbf{w}(i) \sum_{t=1}^{n} \widehat{\mathbf{R}}(t) \cos\left[\frac{\pi}{2n}(2t-1)(i-1)\right], \tag{3}$$

where $i = 1, 2, \ldots, n$ and

$$\mathbf{w}(i) = \begin{cases} \frac{1}{\sqrt{n}}, & i = 1 \\ \sqrt{\frac{2}{n}}, & 2 \leqslant i \leqslant n. \end{cases} \tag{4}$$

15 This process can be reversed to reconstruct the observed signal using the inverse transform

$$\widetilde{\mathbf{R}}(t) = \sum_{i=1}^{n} \mathbf{w}(i)\mathbf{p}(i) \cos\left[\frac{\pi(2t-1)(i-1)}{2n}\right], \tag{5}$$

where $t = 1, 2, \ldots, n$.

### 2.3 Discrete Wavelet Transform

Using the pyramid algorithm, depicted in Fig 1, Mallat (1989) first described the decomposition of an input signal into multi-resolution components using high and low bandpass filters. Each stage of decomposition is referred to as a level. The wavelet 20 decomposition can be performed using a variety of different wavelet families. The most commonly used wavelet family is the Daubechies wavelets (Daubechies, 1990). Each wavelet, within each family, consists of a scaling $\mathrm{h}(m)$ and wavelet $\mathrm{w}(m)$ function, where $m$ is the length along the scaling and wavelet function. The scaling and wavelet functions are used in the low and high pass sequences, respectively. Whilst there are numerous wavelet families that can be chosen for analysis, this study applies the most commonly used Daubechies wavelets. Each choice of wavelet will perform stepwise convolutions of the input




signal over the filter length L. $j_{\max}$ imposes an upper limit on the level of decomposition $j$ that a signal can be decomposed into, where

$$j_{\max} = \left\lfloor \log_2\left( \frac{n+L-1}{2} \right) \right\rfloor, \tag{6}$$

$\lfloor . \rfloor$ is the floor operator and L is the length of the filter being used. The input signal is then convoluted by being passed through high and low pass filters, where

$$p_j(i) = \begin{cases} \sum_{m=1}^{L} \widehat{\mathbf{R}}(2i-m-1)\mathrm{w}(m), & j=1 \\ \sum_{m=1}^{L} \mathbf{p}_{j-1}(2i-m-1)\mathbf{w}(m), & j>1. \end{cases} \tag{7}$$

is the low pass and

$$p_j^{\mathbf{H}}(i) = \begin{cases} \sum_{m=1}^{L} \widehat{\mathbf{R}}(2i-m-1)\mathrm{h}(m), & j=1 \\ \sum_{m=1}^{\mathbf{L}} \mathbf{p}_{j-1}(2i-m-1)\mathrm{h}(m), & j>1. \end{cases} \tag{8}$$

is the high pass, $i = 1,\ldots,n_{j-1}+L-1$ and refers to the $i$th parameter, $j = 1,\ldots,j_{\max}$ and refers to the $j$th level, $m$ refers to the $m$th filter coefficient and $n_{j-1}$ is the length of the input time series. The resultant low pass $\mathbf{p}_j(i)$ and high pass $\mathbf{p}_j^{\mathrm{H}}(i)$ parameters are commonly referred to as approximation and detail parameters, respectively. After the input signal is passed through the high and low pass filters there is an issue of redundancy that needs to be dealt with. The filters split the input

signal into high and low frequency components that each contain roughly half the information of the input signal. As the length of each of the resultant approximation and detail parameter series is equivalent to the length of the input signal, each of the parameter series must be down sampled. The process of down sampling removes every other parameter. It is the process of high and low pass filtering followed by down sampling that enables the DWT to analyze multi-resolution components of a signal. After down sampling the length of the resultant approximation and detail parameter series is

$$n_j = \begin{cases} \left\lfloor \frac{n+L-1}{2} \right\rfloor, & j=1 \\ \left\lfloor \frac{n_{j-1}+\mathbf{L}-1}{2} \right\rfloor, & j>1. \end{cases} \tag{9}$$

where $n_j$ refers to the length of the series at the $j$th level. If further decomposition is required the downsampled low pass may be fed back into the filters until the resultant parameters can no longer be split any further. An iteration of this process is shown in Fig 1. To reverse the decomposition process and reconstruct a signal, up sampling is performed on the parameter series before the lower level parameters are obtained through

$$\mathbf{p}_{j-1}(i) = \sum_{m=\lceil i/2 \rceil}^{\lfloor (L-1+i)/2 \rfloor} \left( \mathbf{p}_j^{\mathrm{H}}(i)\mathrm{h}(2m-i) \right)\left( \mathbf{p}_j(i)\mathrm{w}(2m-i) \right), \quad j>1, \tag{10}$$

where $\lceil . \rceil$ is the ceiling operator and the input signal is reconstructed using

$$\widetilde{\mathbf{R}}(i) = \sum_{m=\lceil i/2 \rceil}^{\lfloor (L-1+i)/2 \rfloor} \left( \mathbf{p}_j^{\mathrm{H}}(i)\mathbf{h}(2m-i) \right)\left( \mathbf{p}_j(i)w(2m-i) \right), \quad j=1, \tag{11}$$





## 3 Data Set

This study utilizes data from the MOPEX data set. Ten years of rainfall data spanning the 1990's for 438 catchments in the United States of America (USA) are used to compare the suitability of the DWT and DCT to represent rainfall time series. Rainfall for the Leaf River catchment (Collins, Mississipi), a catchment that is frequently used for hydrological studies (Sivakumar, 2001; Tang et al., 2006; Bulygina and Gupta, 2011), is used to compare the DWT and DCTs ability to reconstruct high magnitude rainfall events. A single rainfall product for each catchment is used for analysis at a daily time step. A complete description of the selection process and MOPEX data set is given by Schaake et al. (2006). To summarize, only catchments with sufficient rainfall gauge density were selected. Also, the catchments were selected to represent a range of intermediate scale (500 - 10 000 km$^2$) river basins for a range of climates.

## 4 Experiment Design

A major use of both the DWT and DCTs has been in image compression, consequently the observed input signals were compressed and decompressed using a methodology similar to that used in image compression. In order to determine which transform's parameters are able to effectively store the most hydrological input data, both DWT and DCT parameters will be compressed to varying extents for the MOPEX rainfall time series. Before any compression is applied, the original rainfall signal is transformed into DCT and DWT parameters using the processes described in sections 2.2 and 2.3. Each transform is compressed by iteratively zeroing out parameters that provide a low degree of information, these parameters are those closest to zero. A threshold value $T$ [mm] applies a lower limit, for which transform parameters above the threshold are retained. This threshold is iteratively increased until the compressed transform is composed of the desired number of remaining parameters $k$ and percent of Original Parameters POP is met.

$$\text{POP}(T) = 100 \cdot \left( \frac{k}{n} \right), \tag{12}$$

where $k$ becomes smaller as the threshold $T$ increases and $\lim_{T \to \infty} \text{POP} = 0$.

To provide a meaningful comparison between the DCT and the DWTs ability to reproduce different rainfall time series with an increasing POP, a number of simulation performance summary metrics are used. Moriasi et al. (2007) recommend that a combination of graphical techniques and dimensionless and error index statistics be used for model evaluation. For reasons mainly due to their widespread acceptance by the hydrologic community, Moriasi et al. (2007) recommend the use of the Nash-Sutcliffe Efficiency (NSE) and the Root Mean Square Error (RMSE) to Standard deviation of the observed input signal ($\text{RSR} = \text{RMSE}/\sigma_{obs}$). Once the reconstructed signals are obtained, further comparison with the observed rainfall will be made using the bias summary metric. The variance, kurtosis and skewness of the reconstructed signals will be compared with that of the observed signal. The bias is calculated as $\sum_{t=1}^{n} [\widehat{\mathbf{R}}(t) - \widetilde{\mathbf{R}}(t)]/n$, where $\widehat{\mathbf{R}}(t)$ and $\widetilde{\mathbf{R}}(t)$ are the observed and reconstructed rainfall signals, respectively. The reconstructed variance, kurtosis and skewness are all normalized by the observed input signals variance, kurtosis and skewness respectively. The Peak Error (PE) is the peak streamflow error over the 10 year period. It is used to compare the reconstructed and observed signals for seasonal and flood forecasting situations. The





PE is normalized by the peak height of the observed input signal. Further, the number of rain events missed is computed for each reconstruction by flagging original or reconstructed observations that exhibit no rainfall and observations where either the absolute difference between the reconstructed and original observation is less than 0.01 or the ratio of the reconstructed and original observation is equal to zero or larger than 10. Lastly, reconstructed rainfall using the DCT and DWT will be presented

for the Leaf River catchment to compare each transforms ability to reconstruct high magnitude rainfall events.

## 5   Results

Fig 2 shows the relationships between RSR and the number of transform parameters using the DCT and DWT for three different catchments, Arroyo Chico, Skykomish River and Ohio Brush Creek. These catchments represent the smallest, largest and mean rainfall volumes for the MOPEX data set, resepectively. It is clear that for all but the highest POP the DWT is able

to reconstruct the observed signal with lower RSR than the DCT and that as the rainfall volume increases the RSR decreases. For intermediate POPs the DWT is able to reconstruct the observed signal with significantly better RSR than the DCT. As the POP approaches both 100% and 0% there is little discernible difference between the DCT and DWT reconstructions.

Using the reconstructed DWT and DCT signals with 20 POP and using the observed rainfall signal as a reference, a histogram for the Nash-Sutcliffe Efficiency (NSE) is shown for all catchments in Fig 3. Each frequency count in the histogram

represents a catchment from the MOPEX data set. The reconstructed DWT signals are clearly able to better simulate the observed rainfall signal with all DWT reconstructed rainfall signals scoring higher NSE than the DCT reconstructred rainfall signals. Table 1 shows that as the transforms are compressed and fewer parameters are used in the reconstruction, the mean NSE for the DWT stays much closer to the ideal value of one than the DCT. Further the standard deviation of NSE becomes much larger for the DCT.

Fig 4 compares the RSR for the DCT and DWT using four different POPs. A one to one line is included in all subplots and each point represents a catchment from the data set. If the data points fall above the one to one then for that catchment and POP the DWT is able to reconstruct the input rainfall signal with lower RSR. Again it is found that DWT is always able reconstruct the original signal with lower RSR for all POPs. In a similar fashion to that discussed regarding Fig 2 as the POP approaches 0% the difference between the DWT and DCT reconstructions becomes smaller.

The bias, variance and skewness observed in the reconstructed signals for each catchment are shown in Fig 5 for different POPs. The DWT is able to maintain a smaller bias at different POPs for all of the catchments. As the POP decreases the bias becomes increasingly positive and negative for the DWT and DCT, respectively. The distribution of the bias becomes more dispersed for both the DCT and DWT as the POP decreases. The bias can be seen to be dependent on the transform and POP used as well as the catchment being analyzed. Both the DWT and DCT never reconstruct the observed signal with

greater variance than that of the observed rainfall signal. As the POP decreases the normalized variance for the DCT moves further away from unity than the normalized variance for the DWT. The reduction in normalized variance means that, as the POP decreases, both the DWT and especially the DCT reconstructions will have fewer extreme values, when compared to the observed rainfall. The normalized skewness is a measure of symmetry that describes whether or not the reconstructed signal





is more positively skewed (more than one) or less positively skewed (less than one) than the observed input signal. All of the reconstructed and observed signals had a positive normalized skewness. This indicates that, when compressed both transforms will reconstruct the observed rainfall signal with a greater number of values closer to zero when compared to the observed signal. This does not mean that the total volume will be any lower than the total volume of rainfall observed. This is made

evident by the low bias observed in Fig 5.

The normalized kurtosis and PE for all catchments using different POPs are shown in Fig 6. The measure of kurtosis describes how much the fraction of the distributions variance is explained by extreme deviations. Consequently, a normalized kurtosis value larger than one indicates that the reconstructed signals variance is explained more by extreme deviations than the observed input signal. This is likely to be the result of more rainfall values being reconstructed at the extremities than that

of the observed rainfall series. A value smaller than one indicates that the variance is described less by extreme deviations than the observed input signal. Similarly, this is likely to be the result of fewer rainfall values being reconstructed at the extremities than that of the observed rainfall series. It is worth noting that a reconstructed time series can have the same variance yet different kurtosis than the observed rainfall time series. As the POP decreases, the dispersion of normalized kurtosis and skewness increases, and the normalized kurtosis for the DWT and DCT reconstructions becomes larger and smaller than unity,

respectively. With decreasing POP the normalized PE for the reconstructed DWT signal remains small and relatively consistent when compared to the normalized PE for the reconstructed DCT signal.

## 6   Discussion

Fig 3 shows that the DWT and DCT are able to reconstruct the observed input signals with good efficiency using 20 POP. However, the DWT consistently outperforms the DCT. It is observed in Fig 2 that as the POP decreases from 100% the DWT

is able to reconstruct the input signal with increasingly lower RSR than the DCT, the gap in performance is largest when an approximate POP of 40% is used. From this point onwards, as the POP continues to decrease towards 0%, the gap in RSR fades to zero. It is interesting to note that the DWT perfectly reconstructs the observed rainfall signal with as many parameters as there are rainy days whereas the DCT does not.

As the bias for the DWT is consistently close to zero, the use of the DWT for rainfall input data reduction is likely to be

beneficial for hydrologic studies that have short time steps and involve rainfall as an input. Whilst modification of the DWT parameters may slightly overestimate input rainfall, it is not as significant as the consistent underestimation of input rainfall by the DCT. The diminishing ability of both the DWT and DCT to match the input rainfall signal variance indicates that both transforms smooth out input data towards the mean. This trend is more significant in the DCT than the DWT. The impact of this is that, when used as a technique for input data reduction, the DWT will reconstruct temporal variances better than the

DCT. The increased skewness for the reconstructed DWT signals compared to the observed input signals indicates that there is an increase in low magnitude rainfall events. On the contrary, the decreased normalized skewness for the reconstructed DCT signals indicates that a number of the low magnitude rainfall events are tending to increase towards the mean. The kurtosis results shown in Figure 6 demonstrate that, when compared to the observed input signal, events of extreme deviation explain





more of the variance for the reconstructed DWT and less of the variance for the reconstructed DCT. Consequently, as the nature of the extreme deviations is a critical piece of information, the use of the DCT for model input data reduction for hydrologic studies that have short time steps and involving rainfall as an input is not recommended. It is also seen in Figure 6 that the DCT is more likely to miss peak river height information. Consequently, care needs to be taken when choosing a transform

when total volume is critical. Further, the DCT should not be used for studies involving flood forecasting situations where the accuracy of peak river flow is critical.

Whilst it is important that rain gauges measure high volume rainfall events with accuracy and precision, it is also important that low magnitude rainfall events are recorded. Consequently, when evaluating the merits of the DCT and DWT to reconstruct rainfall it would be prudent to analyze the frequency in which each transform is either unable to reconstruct a rainfall event

or erroneously constructs a rainfall event. Table 2 illustrates that, at times, both transforms will either fail to reconstruct a low magnitude rainfall event or will erroneously construct a rainfall event when there was none observed in the original rainfall time series. In general the DWT outperforms the DCT. The exception to this is at 10 POP. This is a results of the discrete nature of the DWT analysis function as opposed to the continuous analysis function used in the DCT. As the POP decreases towards zero both transforms miss more rainfall events.

Due to rapid increases in rainfall intensity, high magnitude rainfall events tend to have high frequency components. In Fig 7 the gradient of the DCT least squares fit to the observed rainfall is lower than the DWT gradient. This indicates that when compressed, the DCT will smooth out high frequency input detail. The smoothing of high frequency high magnitude rainfall events by the DCT is made evident by the lower slope of the linear least squares fit for the DCT of Leaf River rainfall data compared of the DWT. This shows that the compressed DWT is able to retain more detail for high magnitude rainfall events

than the DCT. Using 20 POP, 730 DWT parameters are able to reconstruct observed rainfall with $\mathrm{RSR} = 0.315$, whereas 730 DCT parameters are able to reconstruct observed rainfall with $\mathrm{RSR} = 0.540$. Fig 7 and Fig 8 shows that the DWT often misses and sometimes smooths out low magnitude rainfall events, the DCT however does reconstruct inaccurate rainfall events at these times. Figure 8 validates prior conclusions that, at lower POPs, the DCT will smooth out and underestimate high magnitudes events whilst the DWT will maintain accuracy and precision.

**7  Conclusions**

Succint descriptions of the DCT and DWT were provided to determine the suitability of each transform to be used as a tool for hydrologic model input data reduction. Due to their different construction, each transform provides different possibilities for use in model input data reduction. Since it is infeasible to estimate all transform parameters, the modeller could choose to estimate high or low frequency parameters of the DCT. This would result in minimal control of the temporal component being

modified. Due to the multilevel decomposition of an input signal into high and low frequency parameters by the DWT, the modeller is able to specify the estimation of both time and frequency components. Hence, portions of the input data record can be targeted for estimation. The use of the DWT as a hydrologic model input data reduction technique allows the modeller more flexible options. A comparison of the DWT and DCTs ability to reconstruct MOPEX rainfall data using standard simulation



performance summary metrics, descriptive statistics, and peak errors was then made and it was found that the DWT is most efficient at preserving high magnitude and transient rainfall events. Thus, it is recommended that the DWT be used as a model input data reduction technique for hydrologic studies that have short time steps and involve rainfall as an input. Considering that the bias for the reconstructed DWT rainfall signal is consistently lower than that of the reconstructed DCT signal and that

the skewness, kurtosis and variance are also closest to the input rainfall signal, it is recommended that the DWT also be used as a model input data reduction technique for hydrologic studies that have long time steps with rainfall as an input.

**Appendix A**

**A1**

*Author contributions.*  Ashley Wright conducted the experimental work, contributed towards the theory and wrote the manuscript. Jeffrey

Walker and David Robertson assisted in the writing process. Valentijn Pauwels contributed towards the theory and assisted in the writing process.

*Acknowledgements.*  The authors would like to extend their gratitude to Jasper Vrugt, the anonymous reviewers as well as the Bureau of Meteorology for their comments and recommendations and the provision of data respectively. This work was supported by the Multi-modal Australian ScienceS Imaging and Visualisation Environment (MASSIVE) (www.massive.org.au), a Monash University Engineering Research

Living Allowance stipend, and a top up scholarship from the Bushfire & Natural Hazards Cooperative Research Centre. Valentijn Pauwels is funded by ARC grant FT130100545.



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



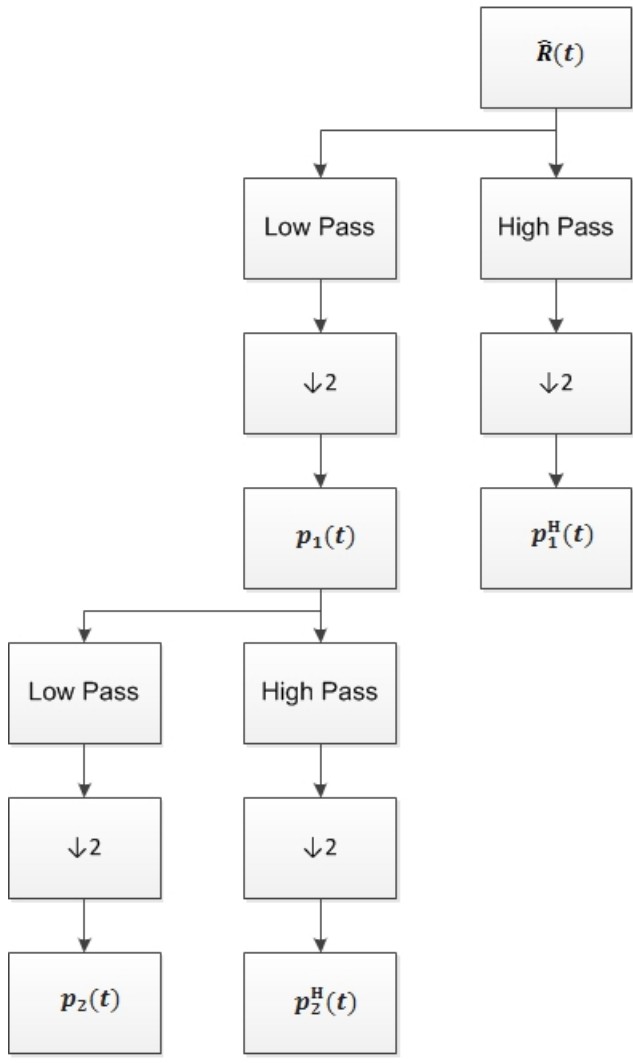

**Figure 1.** A schematic showing the pyramid algorithm used to decompose and down sample ($\downarrow 2$) an input signal into high and low frequency components.

**Table 1.** The mean and standard deviation (StDev) of NSE for the DWT and DCT using a different POP.

|  | NSE DWT | | NSE DCT | |
|---|---|---|---|---|
| POP | Mean | StDev | Mean | StDev |
| 40% | 0.988 | 0.007 | 0.918 | 0.010 |
| 30% | 0.965 | 0.017 | 0.844 | 0.016 |
| 20% | 0.905 | 0.036 | 0.729 | 0.025 |
| 10% | 0.746 | 0.070 | 0.522 | 0.037 |





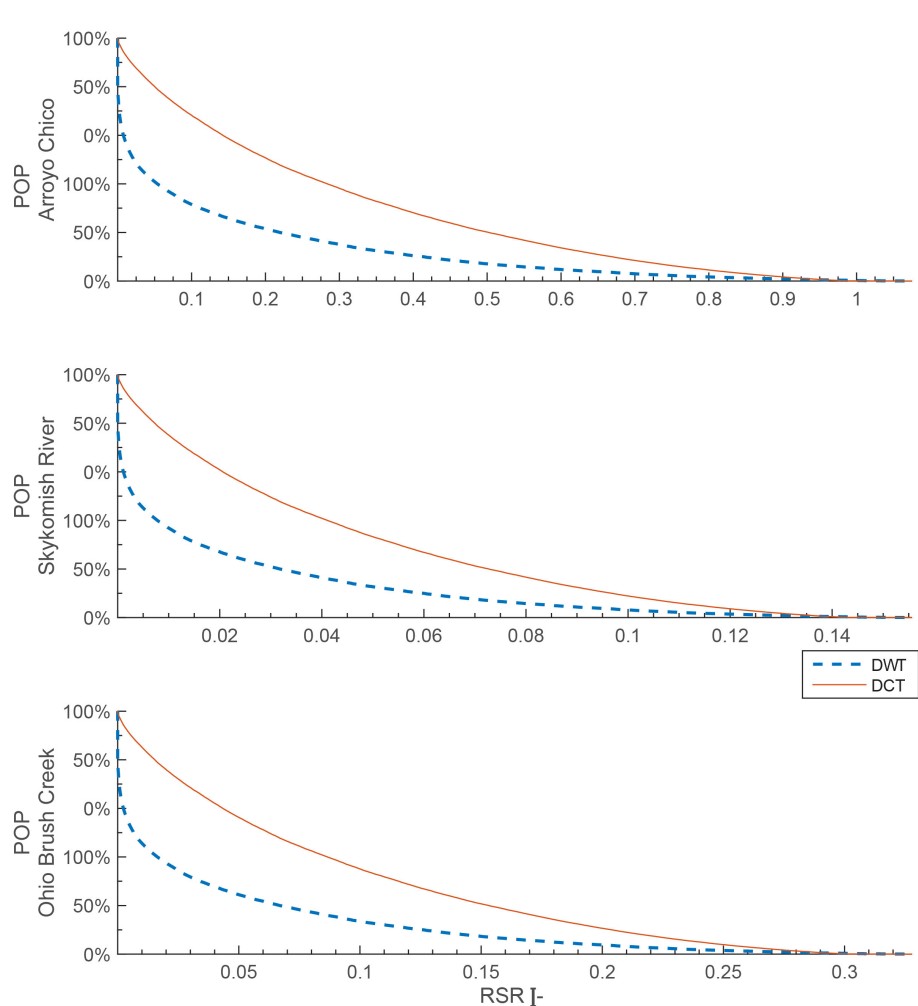

**Figure 2.** Empirical plots showing the relationship between RSR and the POP used for reconstructing an input rainfall signal using the DWT and DCT. The three catchments, from top to bottom of the figure, represent the smallest, largest and mean rainfall volumes throughout the 1990's for the MOPEX data set.





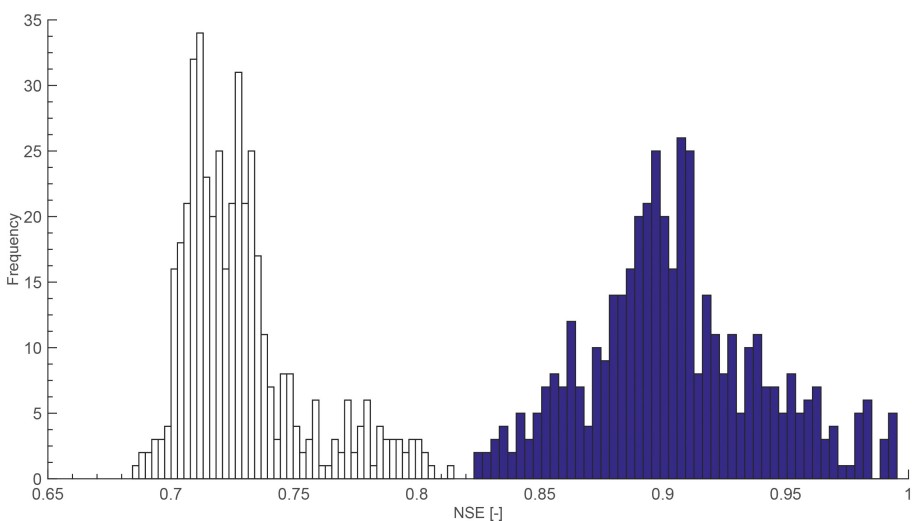

**Figure 3.** Histogram representing the reconstructed DWT (dark bins) and DCT (clear bins) NSE when compared to the observed rainfall signal. Rainfall is reconstructed after the input signal is compressed to 20 POP. Each frequency count represents a catchment from the MOPEX data set.

**Table 2.** The mean and standard deviation (StDev) for the number of missed rainfall events for the DWT and DCT using a different number of parameters.

| | Number of missed rainfall events | | | |
| --- | --- | --- | --- | --- |
| | DWT | | DCT | |
| POP | Mean | StDev | Mean | StDev |
| 40% | 239.004 | 117.317 | 587.934 | 155.375 |
| 30% | 398.005 | 138.793 | 645.495 | 159.378 |
| 20% | 581.591 | 145.769 | 696.288 | 168.524 |
| 10% | 852.340 | 168.590 | 748.075 | 184.910 |





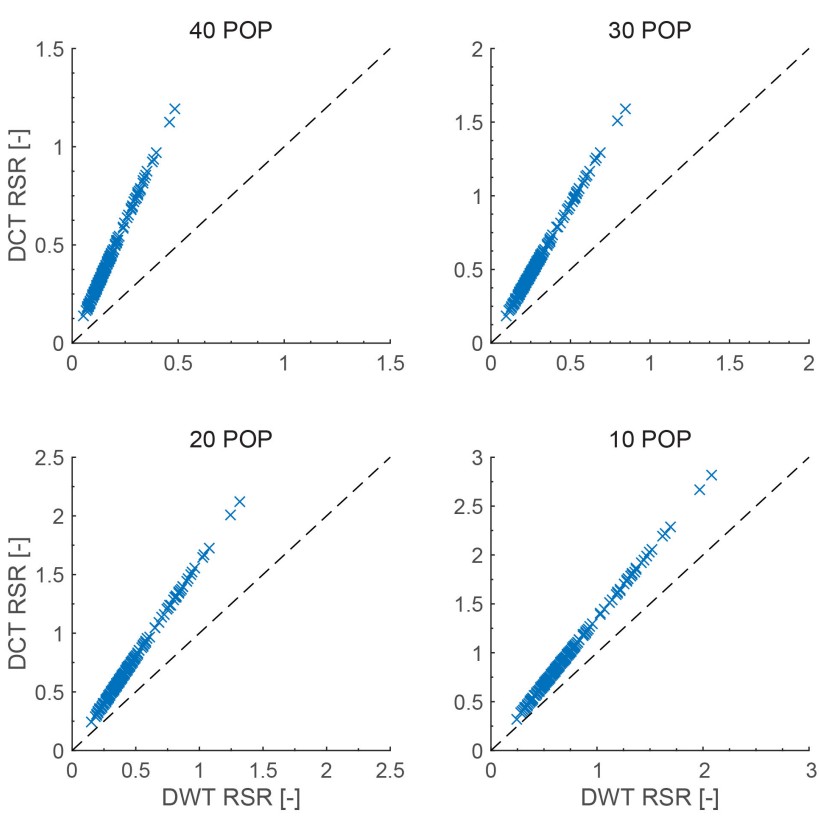

**Figure 4.** Comparative plots of RSR for the DCT and DWT using a different POP. Each data point represents a catchment.





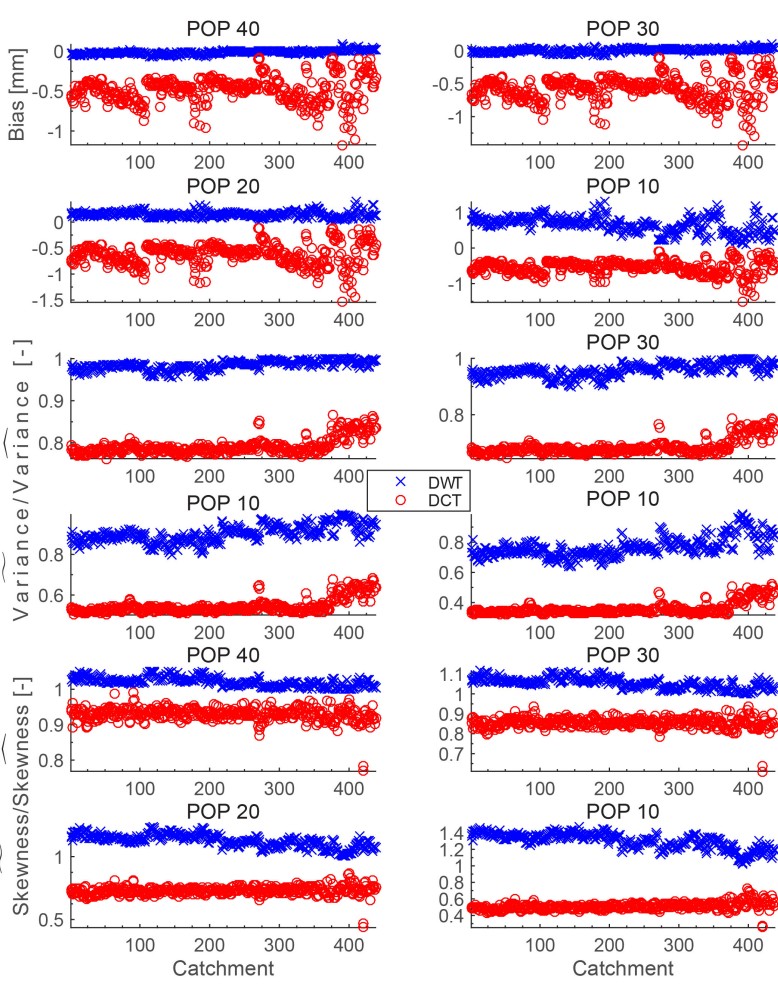

**Figure 5.** Bias and normalized variance and skewness of the reconstructed DWT and DCT signals for each catchment using a different POP.





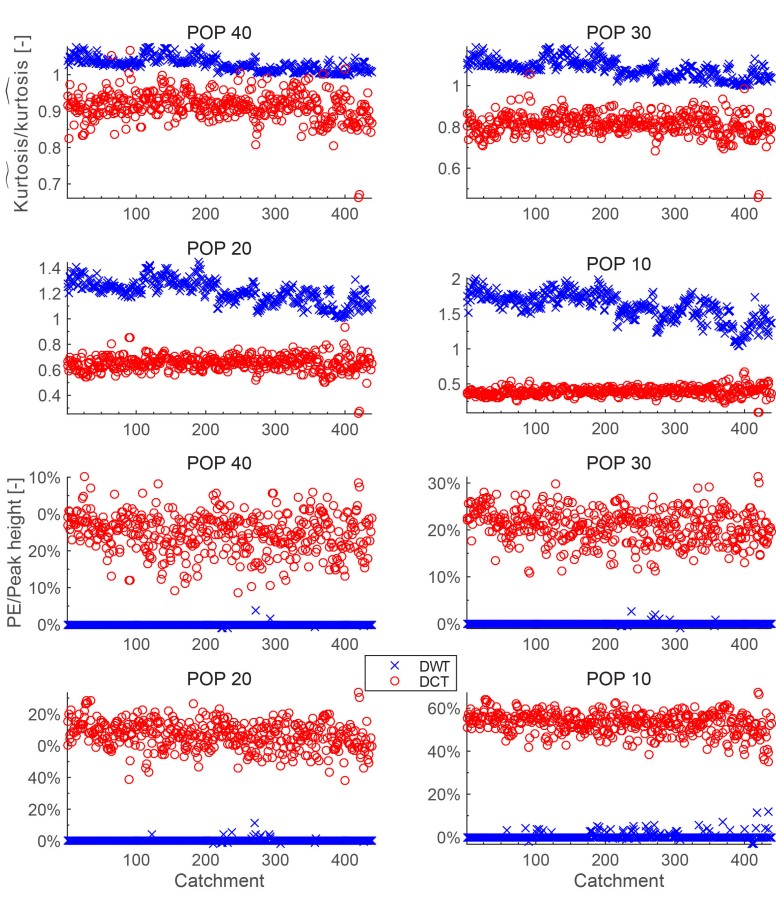

**Figure 6.** Normalized kurtosis of the reconstructed DWT and DCT signals and percentage PE for the reconstructed DWT and DCT signals for each catchment using a different POP.





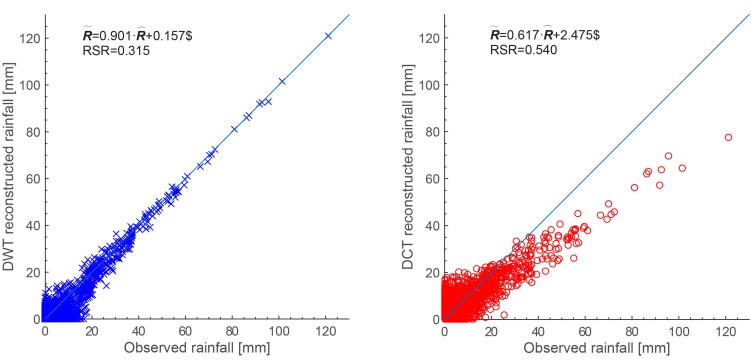

**Figure 7.** Comparison of the reconstructed DCT and DWT signal for the Leaf River (Collins) catchment using 20 POP.





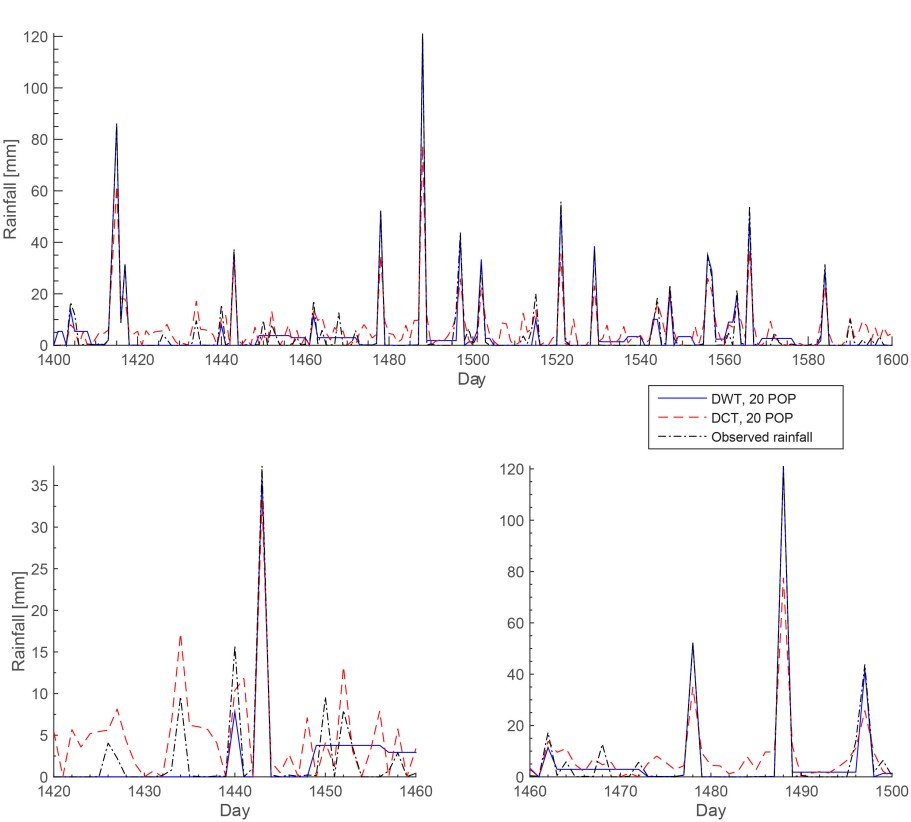

**Figure 8.** The top panel shows a time series comparison of the reconstructed DCT and DWT signal for the Leaf River (Collins) catchment using 20 POP for a period of 200 days. The bottom left and right panels are smaller windows of the same time series during both low and high rainfall periods.