# Peer review of "A Comparison of the Discrete Cosine and Wavelet Transforms for Hydrologic Model Input Data Reduction"

_Hydrology and Earth System Sciences, 2017_

## Referee Comment (RC1) · Anonymous Referee #1 · 14 Mar 2017

Thank you for the chance to review this manuscript. The manuscript is generally well written. However, there are a number of issues that need to be resolved before this manuscript can be accepted for publication.

1. Innovation and contribution of the paper needs to be better defined. The authors compared two methods commonly used in signal processing (i.e. DCT and DWT) for reconstructing rainfall information. But why is this study important? And why these two methods were selected? Are these two methods better than currently used methods? What about other methods used in signal processing, such as short-time Fourier transform (STFT)?

2. Description of experiment design is not very clear. [1] Can you please use a flow

chart to illustrate the steps taken during the experiment? If space is of concern, this reviewer recommends to remove current Figure 1, which is not described in detail and does not have much value. [2] What is the role of stream flow in this study? In the last paragraph on page 7, PE (peak error) is defined as "the peak streamflow error over the 10 year period". Can the authors explain how this error is calculated? How is this error linked to reconstructed rainfall and the performance of the two methods? Are rainfall-runoff models used? If so, these rainfall-runoff models need to be described in the experiment design section. All these information can be included in the flow chart mentioned above, it will help the readers to understand the experiment process. [3] The two methods were not validated – please refer to comment 3.3.

3. Results analysis [1] It is obvious that DWT performs better than DCT from the results obtained. But why is this the case? Is this because the nature of cosine functions oscillating at different frequencies makes DCT unsuitable for rainfall signals that is not cosine in nature? If this is the case, it comes back to my comment 1 above, why is DCT selected for this study at the first place? [2] Figure 4 is a scatter plot of RSR generated using the two methods. It is obvious that they are linearly correlated and the RSR from DWT is always lower than that from DCT. However, what evidence included in this figure show that "DWT is able to reconstruct the input rainfall signal" (line 23 page 8)? [3] The authors claimed that in this study the two methods were "validated" "using several simulation performance summary metrics". Line 24, page 3. This reviewer disagrees. In this study, the performance of the two methods was evaluated using a number of different metrics; however, no independent validation was conducted.

Minor comments: 1) There a few typos throughout the manuscript. For example, Line 7, page 2: "prediction uncertainty" should be "prediction of uncertainty"; Line 22, page 8 "is always able reconstruct" should be "is always able to construct". 2) Line 12 page 1: The sentence does not really make sense here. "Unfortunately, errors in rainfall time series data may lead to hydrological model parameter estimates that produce adequate streamflow simulations during calibration". 3) Figure 1 has only symbols,

which is rather confusing. Please add descriptions in both the figure and caption so the figure stands alone and makes sense.

---

## Author Comment (AC1) · 25 Mar 2017

The authors would like to thank the reviewer for their comments and questions. Please note that all updates to the manuscript will be made once all of the reviews are received. The remarks made by the Reviewer are written in italics, and the replies in normal font.

*Thank you for the chance to review this manuscript. The manuscript is generally well written. However, there are a number of issues that need to be resolved before this manuscript can be accepted for publication. 1. Innovation and contribution of the paper needs to be better defined.*
Thank you for your review of our manuscript. A statement addressing why the study is important is included in the abstract on page 1 Lines 1-4. In short this study is a novel

step towards estimating errors associated with input uncertainty and model structure.

The authors compared two methods commonly used in signal processing (i.e. DCT and DWT) for reconstructing rainfall information. But why is this study important? Further motivation and reasoning why the study is important is addressed in lines 11-23 page 1 and lines 1 -24 on page 2. In summary the use of model input data reduction allows modern parameter estimation algorithms to more efficiently estimate errors associated with input uncertainty and model structure.

*And why these two methods were selected? Are these two methods better than currently used methods?*
A brief reasoning for choosing these two of the many possible transforms is that they are the two most commonly used transforms for model input data reduction techniques in other fields. A more comprehensive reasoning for selecting these two methods is addressed on page 2 lines 23-35 and page 3 lines 1-4. Model Input Data Reduction is currently not performed in hydrology, however the techniques are the most current in other fields.

*What about other methods used in signal processing, such as short-time Fourier transform (STFT)?*
The Discrete Cosine Transform is a version of the STFT. In the manuscript on line 9 page 3 we have stated that the Windowed Fourier Transform is sometimes referred to as the short time Fourier transform. We also mention on line 11 page 5 that the DCT is a version of the WFT.

*2. Description of experiment design is not very clear. [1] Can you please use a flow chart to illustrate the steps taken during the experiment? If space is of concern, this reviewer recommends to remove current Figure 1, which is not described in detail and does not have much value.*
In order to avoid doubling up on presenting the experiment design the authors would prefer not to include a flow chart. However, if it would be deemed necessary, we are certainly willing to provide this. If possible could you please in detail outline which

areas of the experiment design are not clear? We are certainly willing to provide more explanation. The caption of Figure 1 has been expanded upon to provide more detail. This is discussed further in a later comment.

*[2] What is the role of stream flow in this study? In the last paragraph on page 7, PE (peak error) is defined as "the peak streamflow error over the 10 year period". Can the authors explain how this error is calculated? How is this error linked to reconstructed rainfall and the performance of the two methods? Are rainfall-runoff models used? If so, these rainfall-runoff models need to be described in the experiment design section. All these information can be included in the flow chart mentioned above, it will help the readers to understand the experiment process.*

The mention of streamflow is a typographical error. Thank you for pointing out this error. In the manuscript Peak error in fact refers to peak rainfall error.

  *[3] The two methods were not validated – please refer to comment 3.3. 3. Results analysis [1] It is obvious that DWT performs better than DCT from the results obtained. But why is this the case? Is this because the nature of cosine functions oscillating at different frequencies makes DCT unsuitable for rainfall signals that is not cosine in nature? If this is the case, it comes back to my comment 1 above, why is DCT selected for this study at the first place?*

It was clear prior to the study that the DCT would not perform as well as the DWT in reconstructing hydrologic data which are generated by transient mechanisms. This was discussed on page 3 line 15. Yet the use of Fourier transforms remains prevalent in hydrologic studies. Consequently some of the reasons for the selection of the DCT are as follows:

1. To demonstrate that the DCT and Fourier based transforms are not the best transform to use for transformations involving hydrologic rain gauge data,

2. As a baseline from which to compare the DWT to,

3. In the literature it is a commonly used transform for model input data reduction.

*[2] Figure 4 is a scatter plot of RSR generated using the two methods. It is obvious that they are linearly correlated and the RSR from DWT is always lower than that from DCT. However, what evidence included in this figure show that "DWT is able to reconstruct the input rainfall signal" (line 23 page 8)?*

It was never intended for Figure 4 to demonstrate that the DWT is able to reconstruct the input rainfall signal. This information is outlined in Section 2.3. The RSR that is mentioned throughout the text is the RMSE/standard deviation of the reconstructed signal when compared to the observed signal.

*[3] The authors claimed that in this study the two methods were "validated" "using several simulation performance summary metrics". Line 24, page 3. This reviewer disagrees. In this study, the performance of the two methods was evaluated using a number of different metrics; however, no independent validation was conducted.*

Thank you for pointing this out, the wording has been corrected to evaluated.

*Minor comments: 1) There a few typos throughout the manuscript. For example, Line 7, page 2: "prediction uncertainty" should be "prediction of uncertainty"; Line 22, page 8 "is always able reconstruct" should be "is always able to construct".*

The typos in 1) have been corrected.

*2) Line 12 page 1: The sentence does not really make sense here. "Unfortunately, errors in rainfall time series data may lead to hydrological model parameter estimates that produce adequate streamflow simulations during calibration".*

This sentence begins with an explanation of why it is important to address errors in rainfall time series. It is unfortunate that the combination of errors in the input data and parameters often lead to adequate streamflow in the calibration period and poor streamflow in the validation period.

*3) Figure 1 has only symbols, which is rather confusing. Please add descriptions in both the figure and caption so the figure stands alone and makes sense.*

The Figure 1 caption has been updated to provide a stand-alone explanation. And is included below "A schematic showing the pyramid algorithm used to decompose and down sample ($\downarrow 2$) an input signal ($\widehat{\mathbf{R}}$) into high and low frequency components. The

input signal is filtered using the high and low pass filters described in Equations 7 & 8 before being down sampled to produce the level one high and low pass parameters. The low pass parameters are now used as input for the high and low pass filters. This process of filtering and down sampling is repeated until the desired level of decomposition is met."

---

## Referee Comment (RC2) · H. Bazargan (Referee) · 7 May 2017

The paper presents a comparison of two methods of model order reduction, cosine transform and wavelet transform. It is general a smooth good read. However, the contribution of the author is not clear as the similar comparison has been investigated before in fields other than hydrology (please refer to the papers on the model order reduction methods for the modulation schemes on communication channels for example). I think the authors need to clearly state their contribution to this paper. Other than that, the authors have to compare the results with other methods of model reduction, i.e. projection-based methods.

---

## Referee Comment (RC3) · Anonymous Referee #3 · 10 May 2017

**OVERVIEW**

The study investigates the use of Discrete Cosine and Wavelet transforms for the reduction of input data dimensionality in hydrological modelling.

**GENERAL COMMENTS**

I am reviewing the paper after reading the comments raised by previous reviewers on the interactive discussion. As specific comments were already given by previous

reviewers, I included here only my general comments for the paper.

The paper topic seems to be relevant for the HESS readerships. However, I found some important issues that need to be addressed before the publication.

1) It is not clear to me how the DCT and DWT methods are applied. If I well understood, for each basin the authors used streamflow and precipitation data, together with a hydrological model, for applying DCT (and DWT) and thus reducing the dimensionality of precipitation data. However, no hydrological model is mentioned in the paper. What are the input and the output data? Is a hydrological model used? Can the procedure be applied in a testing period? What is the targeted application of the proposed approach (something is mentioned in the introduction, but needs clarifications)? All these questions need to be addressed. Otherwise, I have not clear why the study is relevant for the hydrological community.

2) An important issue in the analysis of rainfall timeseries is related to the zeros, i.e., days with no rainfall. By looking at the results, good performances are obtained for POP values larger than 30-40

3) Besides the performance metrics related to precipitation, also the peak discharge error is mentioned. However, it is not clear how it is computed (see also comment 1). If a hydrological model is used, it should be mentioned. I expect that results depend also on the quality and reliability of discharge time series. If yes, it should be investigated and discussed. All these information are totally missing in the current version of the paper and should be added.

4) Some parts of the paper seem to be written quickly without much attention. Therefore, typos and grammatical errors are present. I suggest a detailed review of the whole

text, and of the figures (e.g., y-axis labels in Figure 2 are wrong).

**RECCOMMENDATION**

On this basis, I found the topic of the paper relevant, but as I mentioned above, the analysis and the text need major revisions before the possible publication on Hydrology and Earth System Sciences.

---

## Author Response (AR1)

The authors would like to thank the reviewer for their comments and questions. The remarks made by the Reviewer are written in italics, and the replies in normal font.

**Reviewer #1**

*Thank you for the chance to review this manuscript. The manuscript is generally well written. However, there are a number of issues that need to be resolved before this manuscript can be accepted for publication. 1. Innovation and contribution of the paper needs to be better defined.*

Thank you for your review of our manuscript. A statement addressing why the study is important is included in the abstract on page 1 Lines 1-4. The introduction has also been reworded to clarify this. To be brief, this study is a novel step towards estimating errors associated with input uncertainty and model structure.

*The authors compared two methods commonly used in signal processing (i.e. DCT and DWT) for reconstructing rainfall information. But why is this study important?*
The introduction has been reworded and restructured to clarify why the study is important. In summary, the use of model input data reduction allows modern parameter estimation algorithms to more efficiently estimate errors associated with input uncertainty and model structure.
*And why these two methods were selected? Are these two methods better than currently used methods?*
A brief reasoning for choosing these two of the many possible transforms is that they are the two most commonly used transforms for model input data reduction techniques in other fields. A more comprehensive reasoning for selecting these two methods is addressed on page 3 lines 4-16. Model Input Data Reduction is currently not performed in hydrology, however the techniques are the most frequently used in other fields.
*What about other methods used in signal processing, such as short-time Fourier transform (STFT)?*

The Discrete Cosine Transform (DCT) is a version of the STFT. In the manuscript on lines 32-33 page 2 we have stated that the Windowed Fourier Transform (WFT) is also sometimes referred to as the STFT. We also mention on line 10 page 5 that the DCT is a version of the WFT.

*2. Description of experiment design is not very clear. [1] Can you please use a flow chart to illustrate the steps taken during the experiment? If space is of concern, this reviewer recommends to remove current Figure 1, which is not described in detail and does not have much value.*

To avoid duplication when presenting the experiment design the authors would prefer not to include a flow chart. However, if it would be deemed necessary, we are certainly willing to provide this. To clarify the steps undertaken, more detail has been added to the Experiment Design section, more specifically on pages 7 lines 16-25.  If further clarification is needed is it possible to outline in detail outline which areas of the experiment design are not clear?  The caption of Figure 1 has been expanded to provide more detail. This is discussed further in a later comment.

*[2] What is the role of stream flow in this study? In the last paragraph on page 7, PE (peak error) is defined as "the peak streamflow error over the 10 year period". Can the authors explain how this error is calculated? How is this error linked to reconstructed rainfall and the performance of the two methods? Are rainfall-runoff models used? If so, these rainfall-runoff models need to be described in the experiment design section. All these information can be included in the flow chart mentioned above, it will help the readers to understand the experiment process.*

The mention of streamflow is a typographical error. Thank you for pointing out this error. In the manuscript Peak error in fact refers to peak rainfall error.

*[3] The two methods were not validated – please refer to comment 3.3. 3. Results analysis [1] It is obvious that DWT performs better than DCT from the results obtained. But why is this the case? Is this because the nature of cosine functions oscillating at different frequencies makes DCT unsuitable for rainfall signals that is not cosine in nature? If this is the case, it comes back to my comment 1 above, why is DCT selected for this study at the first place?*

It was clear prior to the study that the DCT would not perform as well as the DWT in reconstructing hydrologic data which are generated by transient mechanisms. This was discussed on page 3 line 13-15. Yet the use of Fourier transforms remains prevalent in hydrologic studies. Consequently, some of the reasons for the selection of the DCT are as follows:

1. To demonstrate that the DCT and Fourier based transforms are not the best transform to use for transformations involving hydrologic rain gauge data,
2. As a baseline from which to compare the DWT to,
3. In the literature it is a commonly used transform for model input data reduction.

*[2] Figure 4 is a scatter plot of RSR generated using the two methods. It is obvious that they are linearly correlated and the RSR from DWT is always lower than that from DCT. However, what evidence included in this figure show that "DWT is able to reconstruct the input rainfall signal" (line 23 page 8)?*

It was never intended for Figure 4 to demonstrate that the DWT is able to reconstruct the input rainfall signal. This information is outlined in Section 2.3. The RSR that is mentioned throughout the text is the RMSE/standard deviation of the reconstructed signal when compared to the observed signal.

*[3] The authors claimed that in this study the two methods were "validated" "using several simulation performance summary metrics". Line 24, page 3. This reviewer disagrees. In this study, the performance of the two methods was evaluated using a number of different metrics; however, no independent validation was conducted.*

Thank you for pointing this out, the wording has been corrected to evaluate, page 3 line 23, page 10 line 614

*Minor comments: 1) There a few typos throughout the manuscript. For example, Line 7, page 2: "prediction uncertainty" should be "prediction of uncertainty"; Line 22, page 8 "is always able reconstruct" should be "is always able to construct".*

The first instance is not a typo, it is both parameter and prediction uncertainty that the authors refer to. The second typo has been fixed.

*2) Line 12 page 1: The sentence does not really make sense here. "Unfortunately, errors in rainfall time series data may lead to hydrological model parameter estimates that produce adequate streamflow simulations during calibration".*

To provide more clarity this sentence has been rephrased to "Unfortunately, errors in rainfall time series data may lead to hydrological model parameter estimates that produce adequate streamflow simulations only during the calibration period ". Line 12 page 1.

*3) Figure 1 has only symbols, which is rather confusing. Please add descriptions in both the figure and caption so the figure stands alone and makes sense.*

The Figure 1 caption has been updated to provide a stand-alone explanation:

"A schematic showing the pyramid algorithm used to decompose and down sample ($\downarrow 2$) an input signal ($\hat{\mathbf{R}}$) into high and low frequency components. The input signal is filtered using the high and low pass filters described in Equations 7 & 8 before being down sampled to produce the level one high and low pass parameters. The low pass parameters are now used as input for the high and low pass filters. This process of filtering and down sampling is repeated until the desired level of decomposition is met."

**Reviewer #2**

*The paper presents a comparison of two methods of model order reduction, cosine transform and wavelet transform. It is general a smooth good read. However, the contribution of the author is not clear as the similar comparison has been investigated before in fields other than hydrology (please refer to the papers on the model order reduction methods for the modulation schemes on communication channels for example). I think the authors need to clearly state their contribution to this paper. Other than that, the authors have to compare the results with other methods of model reduction, i.e. projection-based methods.*

The manuscript presents a comparison of two methods of model input data reduction, the discrete cosine transform and the discrete wavelet transform, in a hydrological context. The novelty of the paper is that these transforms are discussed in a hydrological context. This is clarified on page 3 line 19-23. Reasons for selecting these two transform are addressed on page 2, lines 29-35 and page 3 lines 1-16. There are a large number of transforms that could be used, the authors chose the two most common transforms that are used for model input data reduction techniques in other fields.

We would also like to clarify that methods of model order reduction are a related field, yet are outside the scope of this paper.

**Reviewer #3**

*OVERVIEW The study investigates the use of Discrete Cosine and Wavelet transforms for the reduction of input data dimensionality in hydrological modelling. GENERAL COMMENTS I am reviewing the paper after reading the comments raised by previous reviewers on the interactive discussion. As specific comments were already given by previous reviewers, I included here only my general comments for the paper. The paper topic seems to be relevant for the HESS readerships. However, I found some important issues that need to be addressed before the publication.*

*1) It is not clear to me how the DCT and DWT methods are applied. If I well understood, for each basin the authors used streamflow and precipitation data, together with a hydrological model, for applying DCT (and DWT) and thus reducing the dimensionality of precipitation data. However, no hydrological model is mentioned in the paper. What are the input and the output data? Is a hydrological model used? Can the procedure be applied in a testing period? What is the targeted application of the proposed approach (something is mentioned in the introduction, but needs clarifications)? All these questions need to be addressed. Otherwise, I have not clear why the study is relevant for the hydrological community.*

We would like to clarify that the authors have only used rainfall data, there is no mention of streamflow data in the Data Set section as streamflow has not been used. The input data is rainfall and the output data is rainfall represented by a smaller number of parameters than the number of rainfall observations. The authors now state that no streamflow data are used on page 7 line 9.

No hydrological model is used, this is now clarified on page 7 line 11.

It is now stated in the Experiment design that there are no calibration of evaluation periods.

To clarify the targeted application, sections of the Abstract, Introduction Model Input Data Reduction Theory sections have been reworded and restructured. In summary, the reduction of model input data allows input data such as rainfall to be reduced to a small number of parameters. Using modern parameter estimation algorithms, the representation of rainfall as parameters allows for the uncertainty in input data to be explored.

If there are specific lines that are unclear, we would like to ask the reviewer to make these known such that modifications can be made to the manuscript. The authors acknowledge that some ambiguity may have arisen due to a typo in which peak streamflow error was mentioned. We apologize for this and would like to clarify that it is peak rainfall error.

*2) An important issue in the analysis of rainfall time series is related to the zeros, i.e., days with no rainfall. By looking at the results, good performances are obtained for POP values larger than 30-40*

The analysis of no rainfall days is indeed important when analysing rainfall time series. As a larger Percentage of Original Parameters (POP) is used, it is expected that the reduced rainfall product correctly represents more days in which no rainfall was observed. Obtaining the best possible representation of rainfall using a minimal number of parameters is a primary concern for model input data reduction. Consequently, it is the reduced rainfall products that have a POP of 40% or less that are of interest to this study.

*3) Besides the performance metrics related to precipitation, also the peak discharge error is mentioned. However, it is not clear how it is computed (see also comment 1). If a hydrological model is used, it should be mentioned. I expect that results depend also on the quality and reliability of discharge time series. If yes, it should be investigated and discussed. All these information are totally missing in the current version of the paper and should be added.*

As was mentioned earlier, the authors made a typo when referring to peak streamflow error. We apologise for any confusion caused. The results shown do not depend on the discharge time series.

*4) Some parts of the paper seem to be written quickly without much attention. Therefore, typos and grammatical errors are present. I suggest a detailed review of the whole Discussion paper text, and of the figures (e.g., y-axis labels in Figure 2 are wrong).*

We have conducted a detailed review of the entire paper and Figure 2 is now amended.

*RECCOMMENDATION On this basis, I found the topic of the paper relevant, but as I mentioned above, the analysis and the text need major revisions before the possible publication on Hydrology and Earth System Sciences.*

[revised manuscript text omitted]

---

## Author Response (AR2)

The authors would like to thank the reviewer and the editor for their comments. The remarks made by the editor are written in italics, and the replies in normal font.

**Editor**

*Page 9, second last and last paragraph (lines 27 and 30) you use "lower" and "smaller". A comparative should be followed by the object it compares with ("lower than…", "smaller than..". It is here clear that the DCT is meant, but I think it would be better to add this.*

Thank you for noticing this. The object now follows the comparative it compares with as suggested. Please see page 8 line 25 and 29.

[revised manuscript text omitted]